# Regularizing the Infinite: Improved Generalization Performance with Deep Equilibrium Models

**Babak Rahmani**            **Jannes Gladrow**            **Kirill Kalinin**

**Heiner Kremer**                        **Christos Gkantsidis**

**Hitesh Ballani**
Microsoft Research, Cambridge, UK CB4 0AB
`{t-brahmani, jannes.gladrow, kkalinin, t-hkremer,`
`christos.gkantsidis, hitesh.ballani}@microsoft.com`

## Abstract

Implicit networks, such as Deep Equilibrium (DEQ) models, present unique opportunities for emerging computing paradigms. Unlike traditional feedforward (FFW) networks, DEQs adaptively adjust their compute resources which has been shown to improve out-of-distribution generalization, especially in algorithmic tasks. We demonstrate that this generalization includes robustness to noise making them well-suited for new hardware, such as analog or optical architectures, with higher yet noisy compute capabilities. But *do DEQ models consistently outperform FFW networks in generalization*? Surprisingly, our findings indicate that this advantage depends heavily on the specific task and network architecture. For equivalent network capacity, DEQ models prove more beneficial as the depth of the network increases—a trend that aligns with hardware systems optimized for deeper networks. We further demonstrate that regularizing the DEQ's entire dynamic process, instead of random initialization or threshold prescribed in previous work, significantly enhances performance across various tasks, including image classification, function regression, adversarial training, and algorithmic extrapolation, making DEQs a compelling choice for next-generation hardware systems.

## 1 Introduction

Deep Equilibrium Models (DEQs) [3] are machine learning models whose predictions result from fixed-points of a learned transformation. By relying on fixed-point iterations these models effectively implement infinite-depth neural networks using only a finite-depth transformation. The DEQ formulation could be beneficial for bi-level optimization, for example in inverse problems [18], and latent space optimization in generative models [14]. Additionally, DEQs are deemed to provide improved performance in out-of-distribution (OOD) generalization compared to traditional feedforward (FFW) thanks to the adaptive inference computation implied by the fixed-point formulation, in contrast to the fixed computation budget of FFWs [21, 2, 16]. To optimize memory usage during training, DEQs apply the implicit function theorem [15], to reduce the memory footprint to $O(1)$ with regards to the number of iterations. Despite memory efficiency, training and inference of DEQs consume more FLOPS compared to FFW networks. A number of methods have been proposed to alleviate the heavy computation of DEQs during training by regularizing the Jacobian of the weights and reducing the number of iterations to convergence [12, 13], approximating the fixed-points [22], or learning fixed-point solvers [5]. Despite all these, DEQs still require convergence of the fixed-point iteration for each input batch of data. This raises the question as to *whether the extra computation compared to traditional FFW architectures provides sufficient benefits*.

38th Second Workshop on Machine Learning with New Compute Paradigms at NeurIPS 2024(MLNPCP 2024).

We demonstrate that generalization capabilities of DEQs extends to the robustness of DEQ models in the presence of noise, which makes them ideal for emerging hardware technologies like analog [27] or optical [29] systems that offer greater compute but tend to be noisier. Counterintuitively, our findings reveal that the advantages of DEQ models over traditional FFWs are not universal; in fact, DEQ models are more robust compared to sufficiently *deep* FFWs. Additionally, we show that introducing noise into the dynamics of the DEQ acts as a regularizer and enhances OOD generalization. We propose modifications to the fixed-point solvers and stopping criteria to enable more efficient fixed-point solving in noisy environments. Our study includes a range of tasks across various domains and problems, from image classification and adversarial robustness to algorithmic reasoning.

## 2    Related work

**Noisy Neural Networks**    Our method is related to prior work that studies the noising of neural network activations, either through dropout or the addition of additive/multiplicative Gaussian noise [23, 10, 28, 8, 19, 17]. These networks can be categorized as either FFWs [8], recurrent neural networks (RNNs) [10, 17], or neural Ordinary Differential Equations (ODEs), which with noise are transformed into stochastic differential equations (SDEs) [28, 19]. It has been demonstrated that noising activations contributes to the improved stability of recurrent networks through explicit [8] or implicit [17] regularization during training. This regularization can promote wider minima in the loss landscape [17, 8] that is advantageous, for example, in adversarial settings [28, 19]. Our work differs from previous work in that it focuses on DEQs with fixed-points, in contrast to SDEs/RNNs that do not necessarily reach a fixed-point. Second, we analytically compare the dynamics of the regularized DEQ and regularized FFW, and study the conditions under which a DEQ is more robust than a FFW. We look at the applications in multiple domains, including regression, classification, algorithmic tasks, and adversarial robustness. In the adversarial setting, we propose to regularize the entire dynamics of the DEQ instead of using costly optimization on the input data [30, 31].

**Out-of-Distribution generalization in DEQs**    Our work is related to the adaptive inference computation of recurrent networks [21] and DEQs [2] to achieve OOD generalization [16]. In the literature, the tasks assigned to DEQ networks are predominantly algorithmic reasoning, such as computing prefix sums and solving 2D mazes [2, 21, 25]. Our work differs in that these studies employ vanilla DEQ models without any form of regularization. We demonstrate that DEQs are not intrinsically superior to FFWs, however, in OOD settings, they tend to perform better under noise when the strength of the distribution shift increases.

## 3    Methods

### 3.1    Preliminaries: DEQ's (perturbed) dynamics

The forward pass iteration of a DEQ can be written as

$$\boldsymbol{z}^{[m+1]} = f_\theta(\boldsymbol{z}^{[m]}; \boldsymbol{x}),\ \boldsymbol{z}^{[0]} = \boldsymbol{0}, \quad \text{stop criterion} \quad \frac{\|f_\theta(\boldsymbol{z}^{[M]}; \boldsymbol{x}) - \boldsymbol{z}^{[M]}\|_2}{\|f_\theta(\boldsymbol{z}^{[M]}; \boldsymbol{x})\|_2} < \epsilon, \tag{1}$$

where $\boldsymbol{x} \in \mathbb{R}^l$ is the input vector, $\boldsymbol{z}^{[m]} \in \mathbb{R}^d$ represents the intermediate state after the $m$-th iteration (with $m = 0, \ldots, M-1$), $f_\theta : \mathbb{R}^d \times \mathbb{R}^l \to \mathbb{R}^d$ defines the transformation at each layer, $\theta$ denotes the set of parameters across the layers, $\|\cdot\|_2$ denotes $l_2$ norm, and $\epsilon$ is a small constant. To find the fixed-point $\boldsymbol{z}^*$ of Eq. 1, solvers such as Broyden's method [7] and Anderson acceleration [1] are employed. These solvers find the solution $\boldsymbol{z}^*$ when either the stop condition in Eq. 1 is met or the threshold $M$ is reached. For distribution shifts in data, which can be formulated as $\boldsymbol{x} + \Delta\boldsymbol{x}$, the dynamics of the DEQs change via:

$$\begin{aligned}
\|\boldsymbol{z}^{[m+1]} - \hat{\boldsymbol{z}}^{[m+1]}\| &= \|f_\theta(\boldsymbol{z}^{[m]}, \boldsymbol{x}) - f_\theta(\hat{\boldsymbol{z}}^{[m]}, \boldsymbol{x} + \Delta\boldsymbol{x})\| \\
&\leq \|f_\theta(\boldsymbol{z}^{[m]}, \boldsymbol{x}) - f_\theta(\hat{\boldsymbol{z}}^{[m]}, \boldsymbol{x})\| + \|f_\theta(\hat{\boldsymbol{z}}^{[m]}, \boldsymbol{x}) - f_\theta(\hat{\boldsymbol{z}}^{[m]}, \boldsymbol{x} + \Delta\boldsymbol{x})\|.
\end{aligned} \tag{2}$$

Here, $\boldsymbol{z}^{[m]}$ represents the state of the unperturbed system after the $m$-th iteration, while $\hat{\boldsymbol{z}}^{[m]}$ denotes the state of the perturbed system at the same iteration. As Eq. (2) suggests, in order to bound the difference between the fixed-point of the model incurred by the original input $\boldsymbol{x}$ and the fixed-point of the distribution-shifted $\boldsymbol{x} + \Delta\boldsymbol{x}$, previous work [31] has utilized an optimization process that corrects perturbed inputs $\boldsymbol{x}$ via a second iterative process at each step of the dynamics to decrease second term in Eq. (2) by reducing the entropy of the perturbed inputs in classification tasks. In contrast, we

propose regularizing the entire dynamics (both terms in Eq. (2)) by introducing random perturbations throughout the evolution of the system. This approach aims to ensure that, for perturbed data, the perturbed and unperturbed dynamics remain closely aligned. This is achieved while being agnostic to the learning task and without requiring a second optimization per step of the DEQ dynamics.

## 3.2 Regularization of DEQ dynamics under noise perturbations

To regularize the dynamics of the DEQ, we introduce perturbations to the intermediate states. Depending on the architecture, this can be achieved by perturbing either the states $z$ or implicitly through perturbing the input injection $x$. Since the magnitude of these states is not known a priori, we choose a multiplicative (input dependent) noise form to ensure we have control over the strength of the signal-to-noise ratio (SNR) of the states. Formally, the states and inputs of $f_\theta$ in Eq. 1 could be perturbed as follows:

$$z^{[m+1]} = f_\theta \left( z^{[m]} + \epsilon_z^{[m]}, x + \epsilon_x \right), \quad z^{[0]} = 0, \tag{3}$$

where

$$\epsilon_z^{[m]} \sim N(\mathbf{0}, \|z^{[m]}\|_2 / \sqrt{\mathrm{SNR}_z}), \quad \epsilon_x \sim N(\mathbf{0}, \|x\|_2 / \sqrt{\mathrm{SNR}_x}). \tag{4}$$

Note that $\mathrm{SNR}_x$ and $\mathrm{SNR}_z$ are hyperparameters chosen to fix the SNR with respect to each variable during the dynamics. Additionally, we consider a stronger case where noise is introduced before the last nonlinearity of the unit-cell. We refer to this as the 'signal' noise with $\mathrm{SNR}_s$. For example for a simple MLP unit-cell $z^{m+1} = \sigma(\boldsymbol{W} z^m + \boldsymbol{x}) + \epsilon_s$, noise $\epsilon_s$ scales with the signal $\boldsymbol{W} z + \boldsymbol{x}$. The fixed-point solution of Eq. 3 is used to minimize the loss function $\mathcal{L}(\theta)$ for learning the task:

$$\min_\theta \mathcal{L} \left( f_\theta(z^* + \epsilon_z^*, \boldsymbol{x} + \epsilon_x) \right), \tag{5}$$

where $z^*$ is the fixed-point solution and $\epsilon_z^*$ is the corresponding noise. To avoid inefficient application of solvers, in the presence of noise, when the fixed-points are already reached, we modify the solvers and stopping criterion to accommodate the variability introduced by noise at each step. We aim to find solutions of the form

$$z = \mathbb{E}_{\epsilon_x, \epsilon_z}[f_\theta(z + \epsilon_z, \boldsymbol{x} + \epsilon_x)], \tag{6}$$

where $z^*$ is the fixed-point of $f_\theta$. Inspired by the stochastic approximation framework, we adopt the Robbins–Monro algorithm [24], which samples the stochastic function $f_\theta$:

$$z^{[m+1]} = (1 + \alpha)z^{[m]} - \alpha f_\theta(z^{[m]} + \epsilon_z, \boldsymbol{x} + \epsilon_x), \quad z^* = \frac{1}{M} \sum_{m=0}^{M-1} z^{[m]}, \tag{7}$$

Here, $z^*$ represents the solution that is the average of the intermediate values up to the $M$-th iteration, while $\alpha$ serves as a hyperparameter. This method is referred to as 'stochastic unrolling.'[1] Rather than averaging all intermediate values, we implement averaging over a window of width $w$ during the solver step of DEQ, which reduces the memory footprint of our model to the size of $w$—often a small number. Additionally, we employ more sophisticated solvers such as Anderson and Broyden. These solvers feature an internal averaging mechanism (e.g., the memory parameter in Anderson), which has the potential to obviate the need for explicit averaging of solutions. As for the stopping criterion, we utilize the relative norm applied to the averaged solution within the window— see appendix B.

## 3.3 Robustness in depth: DEQs are more noise robust than deep FFWs

In this section, we intend to compare regularized DEQ and FFW to determine if a fixed-point network is necessarily associated with stronger robustness. For this we use a simple one-layer fully connected network as the unit-cell for both networks. We assume an additive noise perturbation and refer to the multiplicative perturbation in the Appendix A.1.2 and A.2.2. For DEQ, we have

$$z^{m+1} = \sigma(\boldsymbol{W} z^m + \widetilde{\boldsymbol{x}} + \eta \boldsymbol{\xi}) \tag{8}$$

where $\boldsymbol{W} \in \mathbb{R}^{d \times d}$ is the weight matrix, $\widetilde{\boldsymbol{x}} \in \mathbb{R}^d$ is the input injection vector, $\eta$ is a scalar representing the noise amplitude, $\boldsymbol{\xi} \in \mathbb{R}^d$ is a noise vector with elements drawn from a Gaussian distribution $\mathcal{N}(\mathbf{0}, I)$, and $\sigma$ is the nonlinearity with $\sigma'$ being its derivative. The sensitivity of the DEQ's hidden

---

[1]In the Robbins–Monro algorithm, $\alpha$ is typically inversely proportional to the iteration number $m$. However, empirical evidence suggests that, for a sufficiently large $M$, a fixed $\alpha$ is adequate for finding solutions.

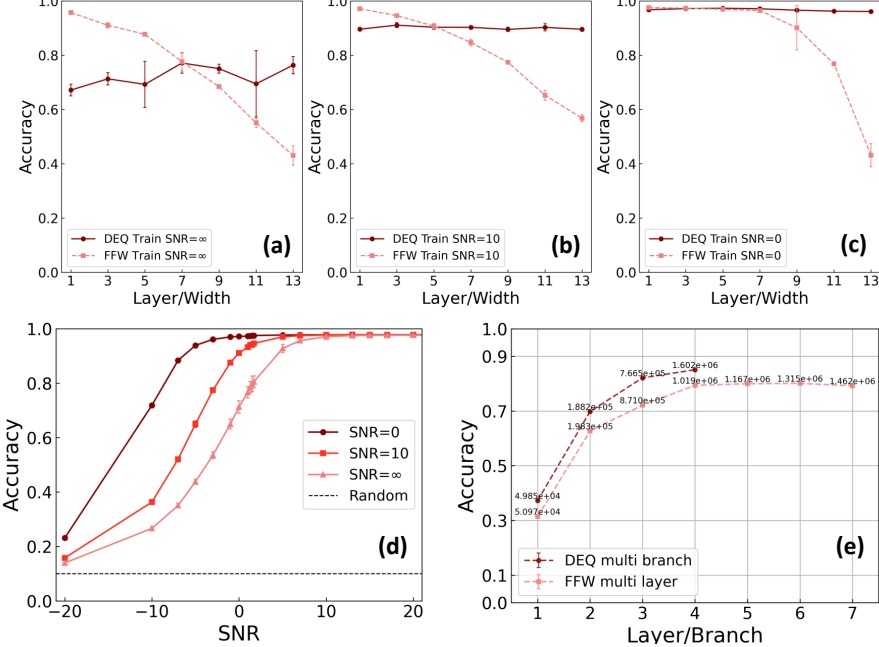

Figure 1: **Robustness of regularized DEQ in classification**: Accuracy of DEQ as a function of increasing width and FFW as a function of increasing network layers on MNIST tested at SNR= 0dB with **(a)** no training noise, **(b)** training noise applied to the dynamics with a signal SNR of 10dB, and **(c)** a signal SNR of 0dB. **(d)** Accuracy of the DEQ in a-c with width= 3 versus inference SNR sweep for three trainig SNRs of 0, 10dB, no noise (SNR= $\infty$). **(e)** Accuracy of MDEQ as a function of increasing branch (scale) and FFW as a function of increasing network layers with an inference signal SNR of 10dB (no train perturbation) on CIFAR-10. The parameter count for networks is indicated above each point.

state with respect to the noise vector $\boldsymbol{\xi}$ can be analyzed through the derivative $\frac{d\boldsymbol{z}^m}{d\xi_i}$, where $\xi_i$ is the $i$-th element of $\boldsymbol{\xi}$. This derivative, at iteration $m+1$, is given by:

$$\frac{d\boldsymbol{z}^{m+1}}{d\xi_i} = \operatorname{diag}(\sigma'_{s_m})(\boldsymbol{W}\frac{d\boldsymbol{z}^m}{d\xi_i} + \eta\boldsymbol{\delta}_i) \tag{9}$$

where $\boldsymbol{s}_m$ is the function argument defined as $\boldsymbol{s}_m := \boldsymbol{W}\boldsymbol{z}^m + \widetilde{\boldsymbol{x}} + \eta\boldsymbol{\xi}$, $\operatorname{diag}(\cdot)$ is the diagonal matrix formed from the vector argument, and $\boldsymbol{\delta}_i$ is the Kronecker delta vector, which has a one at the $i$-th position and zeros elsewhere. The FFW network at layer $m$ is described by the following equation:

$$\boldsymbol{z}^{m+1} = \sigma(\boldsymbol{W}_m\boldsymbol{z}^m + \eta\boldsymbol{\xi}). \tag{10}$$

Here, $\boldsymbol{W}_m \in \mathbb{R}^{d \times d}$ is the weight matrix for layer $m$, and the rest of the terms are as previously defined for the DEQ. The sensitivity of the FFW's hidden state to the noise vector is similarly assessed by $\frac{d\boldsymbol{z}^m}{d\xi_i}$, resulting in:

$$\frac{d\boldsymbol{z}^{m+1}}{d\xi_i} = \operatorname{diag}(\sigma'_{s_m})(\boldsymbol{W}_m\frac{d\boldsymbol{z}^m}{d\xi_i} + \eta\boldsymbol{\delta}_i) \tag{11}$$

To compare the sensitivity of DEQ and FFW to noise, we examine the influence of their respective update equations on the propagation of perturbations to state $\boldsymbol{z}^m$. We note that for the DEQ model, the update Eq. 9 iteratively applies the same weight matrix $\boldsymbol{W}$. The norm of the derivative of the hidden state with respect to the noise, i.e. $\left\|\frac{d\boldsymbol{z}^m}{d\xi_i}\right\|$, is affected by the spectral radius $\rho(\boldsymbol{W})^m$. Conversely, the FFW model's update Eq. 11 involves a product of distinct weight matrices $\boldsymbol{W}_m$. The sensitivity derivative now depends on the spectral radius of the matrix product $\rho(\boldsymbol{W}_1\boldsymbol{W}_2\ldots\boldsymbol{W}_m)$.

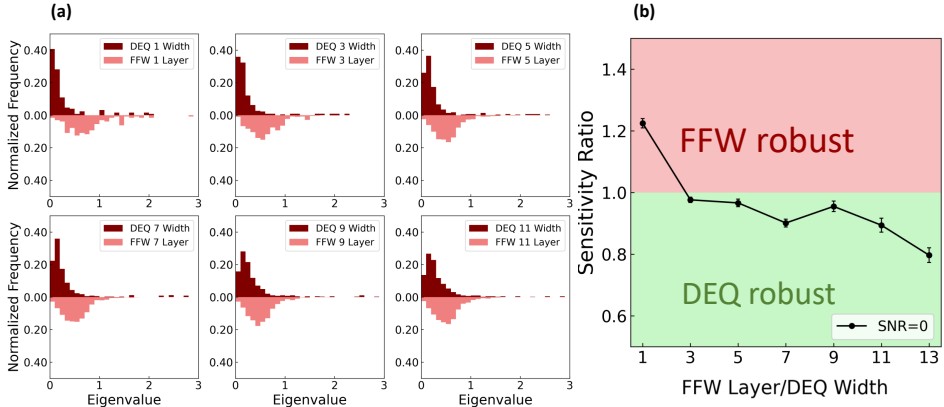

Figure 2: **Sensitivity of DEQ and FFW to noise in classification**: (**a**) Eigen value histogram of DEQ as a function of increasing width and FFW as a function of increasing network layers on MNIST. (**b**) sensitivity ratio of DEQ over FFW. A value smaller than one (green) indicates that the DEQ is less sensitive to the intermediate noise perturbation than the FFW. Models are trained without noise.

In both models, the nonlinearity $\sigma$ plays a crucial role in modulating sensitivity. Assuming a Lipschitz continuous $\sigma$ with Lipschitz constant $L$, the derivative norms could be (loosely) upper bounded by

$$\left\|\frac{d\boldsymbol{z}^m}{d\xi_i}\right\|_{\text{DEQ}} \leq \eta \sum_{k=0}^{m-1} L^{k+1}\rho(\boldsymbol{W})^k, \quad \left\|\frac{d\boldsymbol{z}^m}{d\xi_i}\right\|_{\text{FFW}} \leq \eta \sum_{k=0}^{m-1} L^{k+1}\rho(\boldsymbol{W}_1...\boldsymbol{W}_k). \tag{12}$$

DEQs can be constrained using Jacobian regularization [6] to achieve smaller spectral radii. Similarly, FFWs could also be regularized to have smaller Jacobians. Although this process is more costly for FFWs since the regularization needs to be computed at each layer, as opposed to only at the fixed-point for DEQs, we empirically found that DEQs still maintain smaller Jacobians than their FFW counterparts with Jacobian regularization—see Fig. 13 in Appendix D.2.

## 4  Experiments

We demonstrate the effectiveness of the neural dynamics regularization technique in DEQs to improve OOD generalization in noisy settings across multiple domains in machine learning, including image classification, function regression (see Appendix D.1), adversarial robustness, and algorithmic reasoning (see Appendix D.4). In all domains, we assess whether the fixed-point property of DEQs alone provides robustness advantages over traditional FFW architectures. Our empirical results indicate that DEQs outperform FFWs without any regularization only in deep FFW architectures (Section 3.3). Furthermore, we demonstrate that regularization of DEQs significantly enhances robustness (Section 3.2). We validate DEQs across various architectures, as well as bounded and unbounded nonlinearities to demonstrate generality with respect to architecture. See Appendix E for details of the network architectures and training.

### 4.1  MNIST classification

We begin our examination by assessing the robustness of DEQ models trained on the MNIST dataset against perturbations applied at each iteration of the network. We compare these models to a multilayer FFW model. For the multilayer FFW, we increased the number of layers, each with different weights, while maintaining a consistent hidden size of 128 in each layer. In contrast, for the DEQ models, we expanded the width of the hidden state to ensure the number of parameters remained equivalent between both models. Consequently, the DEQ widths of 128, 223, 288, 340, 386, 427, and 467 correspond to FFWs ranging from single-layer up to thirteen-layer configurations. Figure 1a-c illustrates the accuracy of the DEQ/FFW models against an increasing width/number of layers tested at SNR = 0 dB for different values of training SNRs. Note that for unregularized models (Fig. 1a), DEQ is not robust at low inference SNRs. Yet, when regularized with noise during training, DEQ's performance improves—see Fig. 1d.

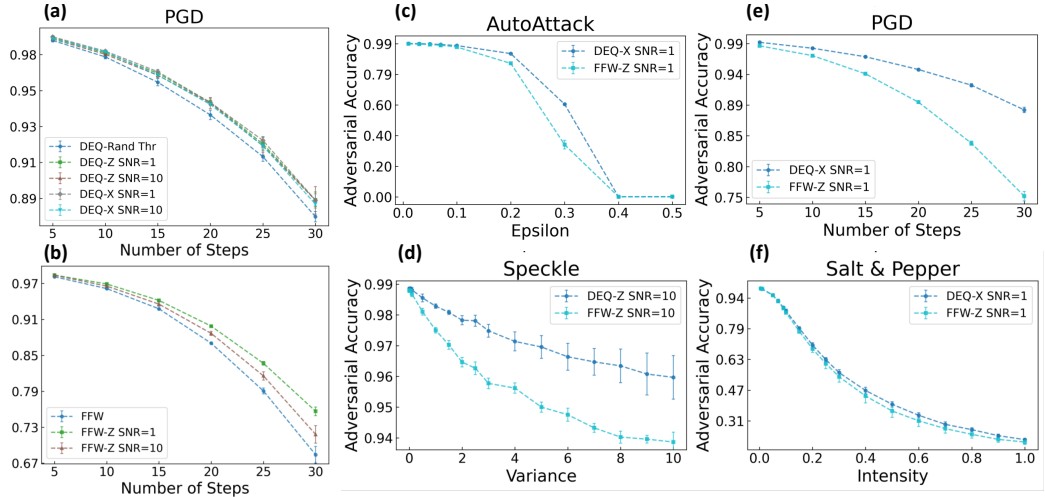

Figure 3: **Improved robustness of DEQ through regularized dynamics in adversarial training**: (**a,b**) Adversarial accuracy of DEQ (top) and FFW (bottom) on MNIST classification, regularized with various SNRs and a random thresholding method, against PGD attacks (step size = 0.1) plotted as versus number of attack steps. (**c-f**) Adversarial accuracy of DEQ and FFW with the strongest defense against various attacks. In each figure, the leftmost part corresponds to weak attacks (clean accuracy) progressing to strong attacks based on parameters specific to each attack.

### 4.1.1 Deep FFWs lag in robustness compared to DEQ

We look into the robustness of the DEQ models in the previous section from the sensitivity perspective. The right hand-side of the Fig. 2 shows the sensitivity ratio $\mathbb{E}_{\xi_i} \left[ \left| \left| \frac{d\boldsymbol{z}^*}{d\xi_i} \right| \right| \right]$ of the DEQ to FFW network, where the sensitivities are averaged over 100 noise samples for all dimensions. Note that $\boldsymbol{z}^*$ represents the fixed-point for the DEQ or the last layer activations for the FFW. We observe that as we increase the depth of the FFW, the DEQ with the corresponding number of parameters shows increasingly superior robustness. We also examine the eigenvalue distribution of the DEQ and FFW models in the left hand-side of Fig. 2. It is noteworthy to note that the DEQ models are less sensitive even when FFW networks are also Jacobian regularized—see Fig. 13 in Appendix D.2. This trend remains consistent as the DEQ models become wider or the FFW models become deeper.

### 4.1.2 CIFAR-10 classification

We conduct experiments on vision tasks using the Multiscale-DEQ (MDEQ) [4], where multiple scales of the image are iterated to a fixed-point. We use up to four scales with number of filters of 32, 64, and 128 for the MDEQ architecture. In contrast, the FFW architecture maintains a consistent number of 128 filters across all layers. The perturbation applied in both DEQ and FFW architectures is a signal perturbation that is added before the nonlinearity of the unit layer. We plot the accuracy of the two architectures versus the branch/layer of the DEQ/FFW for inference perturbation in Figure 1e. These results are consistent with those from the analytical and small-scale DEQ models.

### 4.2 Adversarial robustness

For adversarial experiments, we utilized adversarial training (AT) and dynamics regularization as the defense mechanism. For DEQ models, we propose injecting noise into either $\boldsymbol{z}$ or $\boldsymbol{x}$, while for FFWs, only $\boldsymbol{z}$ is perturbed. Noise is added at two levels of SNRs, 0dB (denoted as SNR 1 in linear unit) and 10dB, with respect to the noise-injected variable. We also considered the random thresholding technique from previous work [31]. In all scenarios, models are trained with either projected gradient descent (PGD) [20] or TRADES [32] for AT with $\ell_\infty$-norm perturbations, $\epsilon = 0.3$ (the maximum allowable perturbation under the $l_\infty$ norm), step size 0.1, and 10 iterations for the PGD training attack. See Appendix E.3 and C for additional details on architecture, training and solver and loss construction in adversarial stetting. Inference-time attacks include white-box attacks such as PGD and Autoattack (AA) [9], as well as black-box attacks like Salt-and-Pepper and Speckle attacks. We

use a convolutional architecture for both DEQ and FFWs (see Appendix E.3 for details). We test adversarial attacks on the MNIST dataset. For each attack, we first identify the strongest defense by understanding which noising regularization $x$, $z$, or thresholding defense, and what SNR provides the strongest attack. Figure 3a,b shows the accuracy of the models for DEQ and FFW under PGD attack and various regularization defenses. As seen, DEQ and SNR$= 1, x$ as well as FFW and SNR$= 1$ usually have the strongest defense. We then use these models and compare the performances of the models against various attacks in Fig. 3c-f and supplementary Fig. 16. Interestingly, as the number of steps of the PGD attack increases, the gap between DEQ and FFW increases. This is also consistent with the strongest AA attack, in which the number of iterations is chosen automatically.

## 5   Broader Impact

In this work, we examined the robustness of DEQ networks, which learn fixed-point representations of data. We demonstrate that the fixed points of DEQs do not inherently offer improved performance compared to FFW networks. Previous work [2] has observed that DEQs mainly exhibit advantages during inference, particularly when applied to OOD algorithmic tasks. However, in the presence of noise, this advantage does not readily extend to common tasks such as classification and regression and is only noticeable for deeper FFW networks. We hypothesize that this behavior can be explained by the spectral norm of both the FFW and DEQ, at least for simple MLP architectures, which is empirically confirmed in larger MDEQ models. Additionally, we have developed a perturbation regularization approach that is task-agnostic and has proven effective in enhancing robustness over FFWs, including in shallow configurations. All these properties make DEQs a strong candidate for emerging analog hardware as a new computing paradigm [29, 27] that is not only robust to the system's inherent noise but also presents an opportunity to provide new capabilities.

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

# A  Noise sensitivity analysis in DEQ and FFW neural networks

Our aim is to understand the robustness of the DEQ/FFW neural networks against intermediate perturbation applied at each iteration/layer of the network.

**Note**: Throughout this section, vectors are denoted by boldface lowercase **x** letters, matrices are indicated by uppercase and boldface letters **X**, and scalar values are represented by non-bold, lowercase letters $x$.

## A.1  DEQ unit-cell

We study two cases of additive and multiplicative perturbations where the perturbation is a normal Gaussian noise.

### A.1.1  Additive Gaussian noise

We are assuming a unit-cell of the following form for the DEQ:

$$z^{m+1} = \sigma(\boldsymbol{W}\boldsymbol{z}^m + \widetilde{\boldsymbol{x}} + \eta\boldsymbol{\xi}) \tag{13}$$

In this equation, $\boldsymbol{z}^m \in \mathbb{R}^d$, $\boldsymbol{W} \in \mathbb{R}^{d \times d}$ represents the weight matrix, $\widetilde{\boldsymbol{x}} \in \mathbb{R}^l$ where $\widetilde{\boldsymbol{x}} = \sigma_2(\boldsymbol{U}\boldsymbol{x})$ is the input projection layer, $\eta$ is a scalar, and $\boldsymbol{\xi}$ is to denote an $d$-dimension noise vector whose elements are from a Gaussian distribution $\mathcal{N}(0, 1)$, and $\sigma$ is the nonlinearity of the unit-cell.

Lets define the pre-nonlinearity activation, i.e. $\boldsymbol{s}_m := \boldsymbol{W}\boldsymbol{z}^m + \widetilde{\boldsymbol{x}} + \eta\boldsymbol{\xi}$. We assume $\boldsymbol{\xi}$ is sampled once and remains fixed throughout the evolution of the equation.

To characterize the robustness of the DEQ, following [26], we look at the vector $\frac{d\boldsymbol{z}^m}{d\xi_i}$ where $\xi_i$ is the $i$-th element of $\boldsymbol{\xi}$. From 13, we have:

$$\frac{d\boldsymbol{z}^{m+1}}{d\xi_i} = (\boldsymbol{W}\frac{d\boldsymbol{z}^m}{d\xi_i} + \eta\boldsymbol{\delta}_i) \circ \sigma'_{s_m} \tag{14}$$

$$= \mathrm{diag}(\sigma'_{s_m})(\boldsymbol{W}\frac{d\boldsymbol{z}^m}{d\xi_i} + \eta\boldsymbol{\delta}_i) \tag{15}$$

where $\boldsymbol{\delta}_i \in \mathbb{R}^d$ represent the Kronecker delta vector, defined as $\boldsymbol{\delta}_i = [0, \ldots, 0, 1_i, 0, \ldots, 0]^t$ where $1_i$ is the unit element positioned at the $i$-th component, and all other elements are zero, $\circ$ is the element-wise hadamard product and $\sigma'_{s_m}$ is the derivative of $\sigma$ computed at $s_m$.

Using Eq. 15, a number of iterations can be written as:

$$\frac{d\boldsymbol{z}^0}{d\xi_i} = 0 \tag{16}$$

$$\frac{d\boldsymbol{z}^1}{d\xi_i} = \eta\left[\mathrm{diag}(\sigma'_{s_0})\boldsymbol{\delta}_i\right] \tag{17}$$

$$\frac{d\boldsymbol{z}^2}{d\xi_i} = \eta\left[\mathrm{diag}(\sigma'_{s_1} \circ \sigma'_{s_0})\boldsymbol{W}\boldsymbol{\delta}_i + \mathrm{diag}(\sigma'_{s_1})\boldsymbol{\delta}_i\right] \tag{18}$$

$$\frac{d\boldsymbol{z}^3}{d\xi_i} = \eta\left[\mathrm{diag}(\sigma'_{s_2} \circ \sigma'_{s_1} \circ \sigma'_{s_0})\boldsymbol{W}\boldsymbol{W}\boldsymbol{\delta}_i + \mathrm{diag}(\sigma'_{s_2} \circ \sigma'_{s_1})\boldsymbol{W}\boldsymbol{\delta}_i + \mathrm{diag}(\sigma'_{s_2})\boldsymbol{\delta}_i\right] \tag{19}$$

$$\ldots \tag{20}$$

$$\frac{d\boldsymbol{z}^m}{d\xi_i} = \eta\sum_{k=0}^{t-1}\mathrm{diag}(\prod_{j=0}^{k}\sigma'_{s_{m-1-j}})\boldsymbol{W}^k\boldsymbol{\delta}_i \tag{21}$$

where $\boldsymbol{W}^k$ denotes $k$ application of matrix $\boldsymbol{W}$, i.e. $\boldsymbol{W}^k := \underbrace{\boldsymbol{W}\boldsymbol{W}\cdots\boldsymbol{W}}_{k \text{ times}}$.

### A.1.2 Multiplicative Gaussian noise

We are assuming a unit-cell of the following form for the DEQ:

$$z^{m+1} = \sigma(Wz^m + \widetilde{x} + G_m(z^m, \widetilde{x})\xi) \tag{22}$$

where

$$G_m(z^m, \widetilde{x}) = ||Wz^m + \widetilde{x}||_2 \tag{23}$$

Similar to the additive noise, the sensitivity parameter $\frac{dz^m}{d\xi_i}$ reads as:

$$\frac{dz^{m+1}}{d\xi_i} = (W\frac{dz^m}{d\xi_i} + G_m\delta_i + \frac{dG_m}{d\xi_i}\xi) \circ \sigma'_{s_m} \tag{24}$$

$$= \text{diag}(\sigma'_{s_m})(W\frac{dz^m}{d\xi_i} + G_m\delta_i + \frac{dG_m}{d\xi_i}\xi) \tag{25}$$

In Eq. 25, $\frac{dG_m}{d\xi_i}$ can be further simplified using 23:

$$\frac{dG_m}{d\xi_i} = \frac{(Wz^m + \widetilde{x})^t(W\frac{dz^m}{d\xi_i})}{||Wz^m + \widetilde{x}||_2} = \frac{(Wz^m + \widetilde{x})^t(W\frac{dz^m}{d\xi_i})}{G_m} = \widetilde{s}_m^t W\frac{dz^m}{d\xi_i} \tag{26}$$

where $\widetilde{s}_m^t$ is the normalized $s_m$. Using Eqs. 25 and 26, a number of iterations can be written as:

$$\frac{dz^0}{d\xi_i} = 0 \tag{27}$$

$$\frac{dz^1}{d\xi_i} = \left[G_0\text{diag}(\sigma'_{s_0})\delta_i\right] \tag{28}$$

$$\frac{dz^2}{d\xi_i} = \left[G_1\text{diag}(\sigma'_{s_1})\delta_i + G_0\text{diag}(\sigma'_{s_1} \circ \sigma'_{s_0})W\delta_i + G_0\text{diag}(\sigma'_{s_1})s_1^t W\delta_i\xi\right] \tag{29}$$

$$... \tag{30}$$

$$\frac{dz^m}{d\xi_i} = \left[\sum_{k=0}^{t-1}\left(G_k\text{diag}\left(\prod_{j=k+1}^{t-1}\sigma'_{s_j}\right)W^{t-1-k}\delta_i\right)\right] \tag{31}$$

### A.2 FFW unit-cell

We study two cases of additive and multiplicative perturbations where the perturbation is a normal Gaussian noise.

### A.2.1 Additive Gaussian noise

We are assuming hidden layers which are of the following form for the FFW network:

$$z^{m+1} = \sigma(W_m z^m + \eta\xi) \tag{32}$$

where $z^0 = \widetilde{x} = \sigma_2(Ux)$, and $W_i$ represents the weight matrix at the $i$-th hidden layer.

To characterize the robustness of the FFW model, we again look at the vector $\frac{dz^m}{d\xi_i}$. From 32, we have:

$$\frac{dz^{m+1}}{d\xi_i} = (W_m\frac{dz^m}{d\xi_i} + \eta\delta_i) \circ \sigma'_{s_m} \tag{33}$$

$$= \text{diag}(\sigma'_{s_m})(W_m\frac{dz^m}{d\xi_i} + \eta\delta_i) \tag{34}$$

Using Eq. 34, a number of iterations can be written as:

$$\frac{d\boldsymbol{z}^0}{d\xi_i} = 0 \tag{35}$$

$$\frac{d\boldsymbol{z}^1}{d\xi_i} = \eta \left[ \text{diag}(\sigma'_{s_0})\boldsymbol{\delta}_i \right] \tag{36}$$

$$\frac{d\boldsymbol{z}^2}{d\xi_i} = \eta \left[ \text{diag}(\sigma'_{s_1} \circ \sigma'_{s_0})\boldsymbol{W}_1\boldsymbol{\delta}_i + \text{diag}(\sigma'_{s_1})\boldsymbol{\delta}_i \right] \tag{37}$$

$$\frac{d\boldsymbol{z}^3}{d\xi_i} = \eta \left[ \text{diag}(\sigma'_{s_2} \circ \sigma'_{s_1} \circ \sigma'_{s_0})\boldsymbol{W}_2\boldsymbol{W}_1\boldsymbol{\delta}_i + \text{diag}(\sigma'_{s_2} \circ \sigma'_{s_1})\boldsymbol{W}_2\boldsymbol{\delta}_i + \text{diag}(\sigma'_{s_2})\boldsymbol{\delta}_i \right] \tag{38}$$

$$\dots \tag{39}$$

$$\frac{d\boldsymbol{z}^m}{d\xi_i} = \eta \sum_{k=0}^{t-1} \text{diag}(\prod_{j=0}^{k} \sigma'_{s_{m-1-j}})\boldsymbol{W}^k\boldsymbol{\delta}_i \tag{40}$$

where $\boldsymbol{W}^k$ denotes $k$ application of matrix $\boldsymbol{W}_k$, i.e. $\boldsymbol{W}^k := \underbrace{\boldsymbol{W}_k\boldsymbol{W}_{k-1} \cdots \boldsymbol{W}_1}_{k \text{ times}}$.

### A.2.2 Multiplicative Gaussian noise

We are assuming hidden layers which are of the following form for the FFW network:

$$\boldsymbol{z}^{m+1} = \sigma(\boldsymbol{W}_m\boldsymbol{z}^m + \widetilde{\boldsymbol{x}} + G_m(\boldsymbol{z}^m, \widetilde{\boldsymbol{x}})\boldsymbol{\xi}) \tag{41}$$

where

$$G_m(\boldsymbol{z}^m, \widetilde{\boldsymbol{x}}) = ||\boldsymbol{W}_m\boldsymbol{z}^m + \widetilde{\boldsymbol{x}}||_2 \tag{42}$$

Similar to the additive noise, the sensitivity parameter $\frac{d\boldsymbol{z}^m}{d\xi_i}$ reads as:

$$\frac{d\boldsymbol{z}^{m+1}}{d\xi_i} = (\boldsymbol{W}_m\frac{d\boldsymbol{z}^m}{d\xi_i} + G_m\boldsymbol{\delta}_i + \frac{dG_m}{d\xi_i}\xi) \circ \sigma'_{s_m} \tag{43}$$

$$= \text{diag}(\sigma'_{s_m})(\boldsymbol{W}_m\frac{d\boldsymbol{z}^m}{d\xi_i} + G_m\boldsymbol{\delta}_i + \frac{dG_m}{d\xi_i}\xi) \tag{44}$$

In Eq. 44, $\frac{dG_m}{d\xi_i}$ can be further simplified using 42:

$$\frac{dG_m}{d\xi_i} = \frac{(\boldsymbol{W}_m\boldsymbol{z}^m + \widetilde{\boldsymbol{x}})^t(\boldsymbol{W}_m\frac{d\boldsymbol{z}^m}{d\xi_i})}{||\boldsymbol{W}\boldsymbol{z}^m + \widetilde{\boldsymbol{x}}||_2} = \frac{(\boldsymbol{W}_m\boldsymbol{z}^m + \widetilde{\boldsymbol{x}})^t(\boldsymbol{W}_m\frac{d\boldsymbol{z}^m}{d\xi_i})}{G_m} = \widetilde{\boldsymbol{s}}_m^t\boldsymbol{W}_m\frac{d\boldsymbol{z}^m}{d\xi_i} \tag{45}$$

where $\widetilde{\boldsymbol{s}}_m^t$ is the normalized $\boldsymbol{s}_m$. Using Eqs. 44 and 45, a number of iterations can be written as:

$$\frac{d\boldsymbol{z}^0}{d\xi_i} = 0 \tag{46}$$

$$\frac{d\boldsymbol{z}^1}{d\xi_i} = \left[ G_0\text{diag}(\sigma'_{s_0})\boldsymbol{\delta}_i \right] \tag{47}$$

$$\frac{d\boldsymbol{z}^2}{d\xi_i} = \left[ G_1\text{diag}(\sigma'_{s_1})\boldsymbol{\delta}_i + G_0\text{diag}(\sigma'_{s_1} \circ \sigma'_{s_0})\boldsymbol{W}_m\boldsymbol{\delta}_i + G_0\text{diag}(\sigma'_{s_1})\boldsymbol{s}_1^t\boldsymbol{W}_m\boldsymbol{\delta}_i\boldsymbol{\xi} \right] \tag{48}$$

$$\dots \tag{49}$$

$$\tag{50}$$

### A.3 Effect of the nonlinearity on the robustness of FFWs

In the case of additive perturbation, we compare the sensitivity of the DEQ to a feed-forward network with one hidden layer. This specific scenario is the only one in which the two networks have the same number of parameters. In this case, we have

$$\frac{d\boldsymbol{z}_{\text{out}}}{d\xi_i} = \eta \left[ \boldsymbol{\delta}_i \circ \sigma'_{s_0} \right], \qquad\qquad [\text{FFW}, s_0 = \boldsymbol{W}_1 \widetilde{\boldsymbol{x}} + \eta \boldsymbol{\xi}] \qquad (51)$$

$$\frac{d\boldsymbol{z}_{\text{out}}}{d\xi_i} = \eta \sum_{k=0}^{M-1} \left( \boldsymbol{W}^k \boldsymbol{\delta}_i \circ \prod_{j=0}^{k} \sigma'_{s_{m-1-j}} \right), \qquad\qquad [\text{DEQ}] \qquad (52)$$

Here, $M$ represents the number of iterations required for the DEQ network to converge. We observe that for a single hidden layer in an FFW, the sensitivity is not directly tied to $\boldsymbol{W}_1$, instead, it is related indirectly through the derivative of the nonlinearity $\sigma'_{s_0}$. Nonlinear functions, such as the ReLU, which have a derivative of zero for certain input ranges, tend to be more robust.

## B Additional Details on the Fixed-Point Solving in the Presence of Noise

We propose two mechanisms that could augment fixed-point solvers used in DEQ models in the presence of noise. One is the averaging of the solutions using stochastic approximation, and the other is modifying the stopping criterion based on the averaged values. In particular, we propose to use a stochastic-unrolling (SU) solver based on

$$\boldsymbol{z}^{[m+1]} = \boldsymbol{z}^{[m]} - \alpha f_\theta(\boldsymbol{z}^{[m]} + \boldsymbol{\epsilon}_z, \boldsymbol{x} + \boldsymbol{\epsilon}_x), \quad \tilde{\boldsymbol{z}}^* = \frac{1}{M} \sum_{i=0}^{M-1} \boldsymbol{z}^{[i]}, \qquad (53)$$

and a relative-averaged stopping criterion over a window of length $w$ defined as:

$$\frac{\|f_\theta(\tilde{\boldsymbol{z}}^{[w]}; \boldsymbol{x}) - \tilde{\boldsymbol{z}}^{[w]}\|_2}{\|f_\theta(\tilde{\boldsymbol{z}}^{[w]}; \boldsymbol{x})\|_2} < \epsilon \qquad (54)$$

where $\tilde{\boldsymbol{z}}^{[w]}$ is to denote averaging over the window $w$. We note that other fixed-point solvers, such as the Anderson accelerator [1] or Broyden's method [7], could utilize the averaged-relative stopping criterion. To compare the performances of the solvers in the presence of noise we use a DEQ with a unit-cell consisting of one hidden layer of size 100 with a fully connected structure and Tanh nonlinearity, where the weight matrix is initialized in $U(-1/\sqrt{100}, 1/\sqrt{100})$. We consider multiplicative perturbation and two regimes of low (0dB) and high (20dB) SNR levels.

Empirically, we find that averaging assists in finding a solution more effectively than the native unrolling solver (see Fig. 4a-d). Methods such as Anderson's and Broyden's can find solutions without explicit averaging due to their inherent averaging capabilities. We observe that the averaged-relative criterion (see Fig. 4c,d) is more effective at detecting the fixed-point compared to the relative criterion (see Fig. 4a,b), potentially preventing unnecessary continuation of the solving process, thereby speeding up the procedure and reducing computation.

To further demonstrate the improvement of the average-relative stopping criterion, we apply it to MNIST classification with signal (pre-activation) noising (see Fig. 5). The average-relative criterion provides a lower number of iterations for convergence. We also examine the impact of the stochastic solver on performance. Using the MDEQ architecture with four branches on CIFAR-10 classification with test-time signal noise perturbation (see Fig. 6), we note that the stochastic solver achieves the same accuracy as the Broyden method.

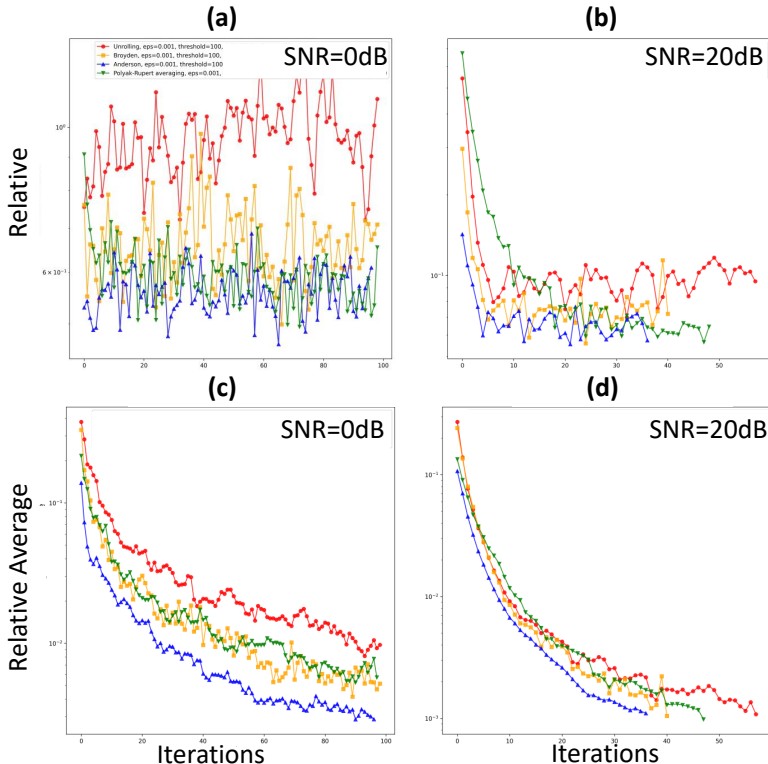

Figure 4: Traces of relative **(a, b)** and average-relative **(c, d)** difference errors of solvers (Anderson, Broyden, unrolling, stochastic unrolling) for a one-layer randomly initialized MLP of size 100 with Tanh nonlinearity when the model state $z$ is perturbed with Gaussian noise in two cases: SNR = 20 dB and SNR = 0 dB.

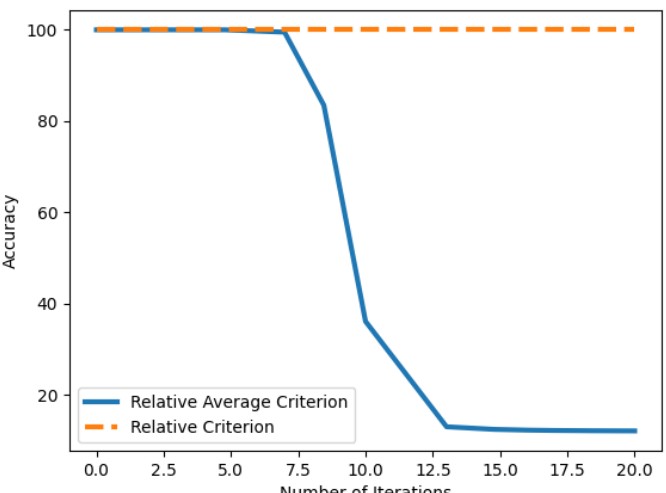

Figure 5: The number of iterations of the unrolling solver with relative and average-relative difference error on MNIST classification for a DEQ model with a single MLP unit-cell architecture of size 128, with pre-activation perturbation of various SNRs. In both cases, $\epsilon = 0.01$.

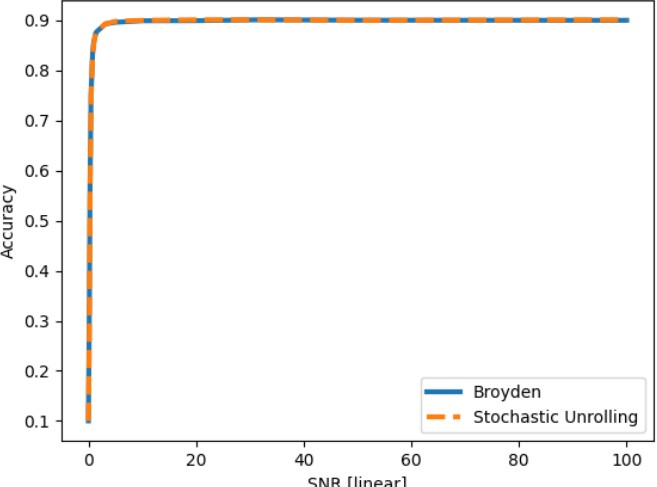

Figure 6: Accuracy of a large-scale MDEQ architecture (4 branches) trained on CIFAR-10 and tested on SNR ranges. Noise is added to the pre-activation signal in a unit cell of the architecture and each branch. Both Broyden and stochastic unrolling provide the same test-time accuracy.

## C  Additional details on the construction of adversarial loss in DEQ

As the gradient tape for the intermediate states of DEQs are not stored by the solvers, loss gradient cannot backpropagate to the input. We followed [31] and employed fixed-point solutions followed by

unrolling for a few steps:

$$z_j^* = (1 - \lambda)z_{j-1}^* + \lambda f_\theta(z_{j-1}^*; x) \tag{55}$$

with $z_j^*$ as the solution of the fixed-point solver and $j$ ranging from $1$ to $K$. The resulting state, $z_K^*$, post-unrolling was then utilized to calculate the loss and gradients as:

$$\min_\theta \max_{\Delta x \in [-\epsilon, \epsilon]^l} \mathcal{L}\left(z_K^*, y\right), \tag{56}$$

where $\Delta x$ represents the input perturbed, and $y$ is the ground-truth label. We used $M = 8$ (unrolling threshold), $K = 9$ and $\lambda = 0.5$.

# D    Additional experiments

Additional experiments for MNIST classification, regression, and Adversarial training is provided.

## D.1    1-Dimensional regression

We investigated the robustness of DEQ models in regression tasks. Specifically, we examined the regression of a 1-D function defined by $y = 2(\exp(-\frac{x^2}{2\sigma^2}) - 0.5))$, where $\sigma = 0.25$. We sampled 10,000 points $x \in [-1, 1]$ and their corresponding $y$ values. We used a multilayer FFW model with a consistent hidden size of 128 across layers with odd number of layers from one up to eleven layers. The corresponding DEQ model has a single layer with increasing hidden sizes of 128, 223, 288, 340, 386, and 427. These sizes correspond to networks with increasing width to match the number of parameters to the FFW network in each case. Both DEQ and FFW have the same unit-cell nonlinearity of Tanh. But for input projection of $x$, we examined three activations: ReLU, SiLU, and identity (no activation). We studied a signal noise perturbation where the signal before activation is perturbed. During training, no perturbation (equivalent to SNR of $\infty$) is applied. At inference, we varied the SNR coefficient to evaluate the model under different SNRs.

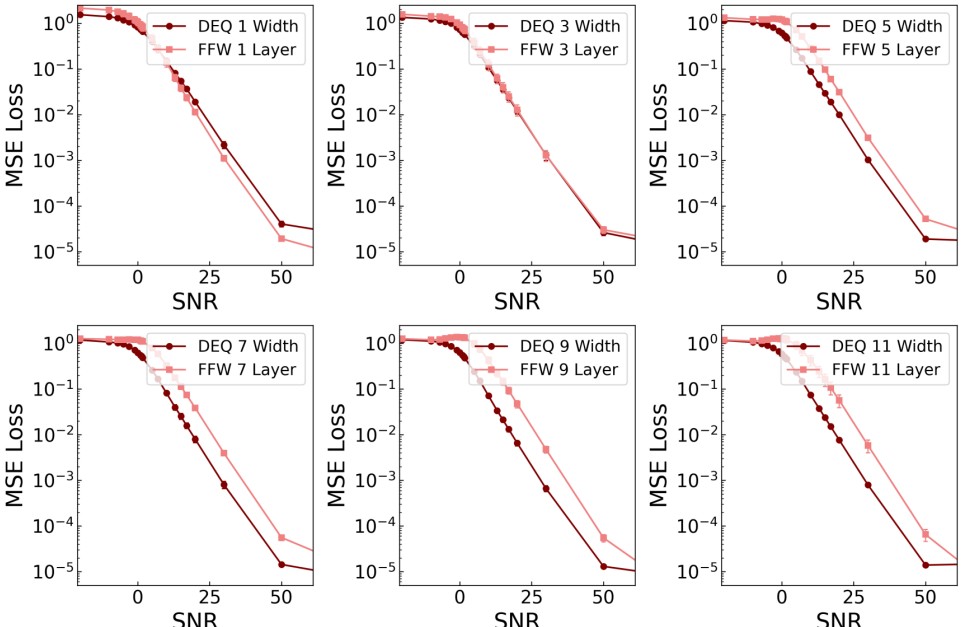

Figure 7: MSE loss for 1-D regression task versus test-time signal SNR in DEQ and FFWs, where the DEQ width is matched to the number of layers in the FFW to equalize the parameters. The input projection layer has no activation.

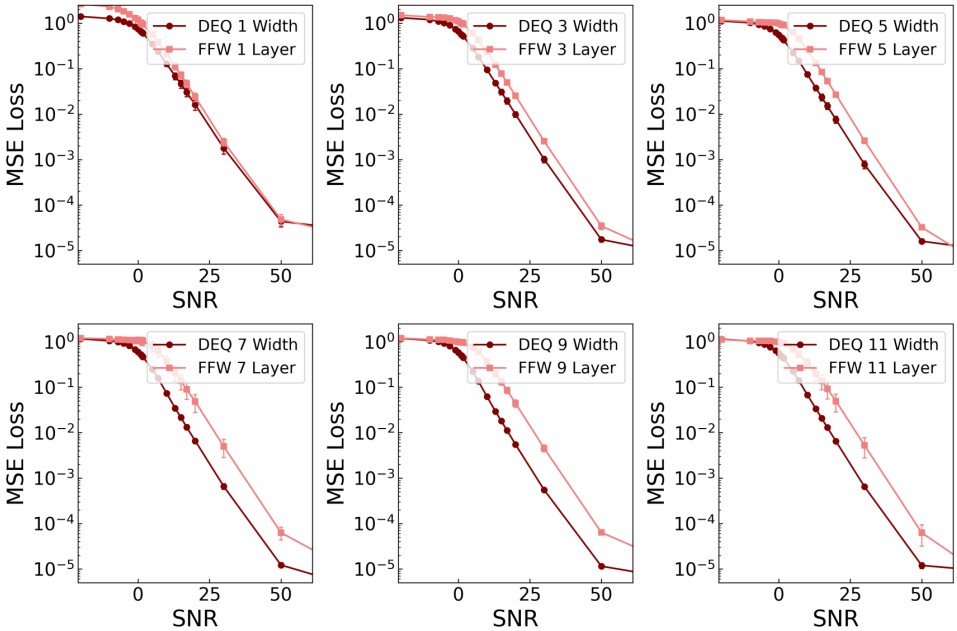

Figure 8: MSE loss for 1-D regression task versus test-time signal SNR in DEQ and FFWs, where the DEQ width is matched to the number of layers in the FFW to equalize the parameters. The input projection layer has a SiLU activation.

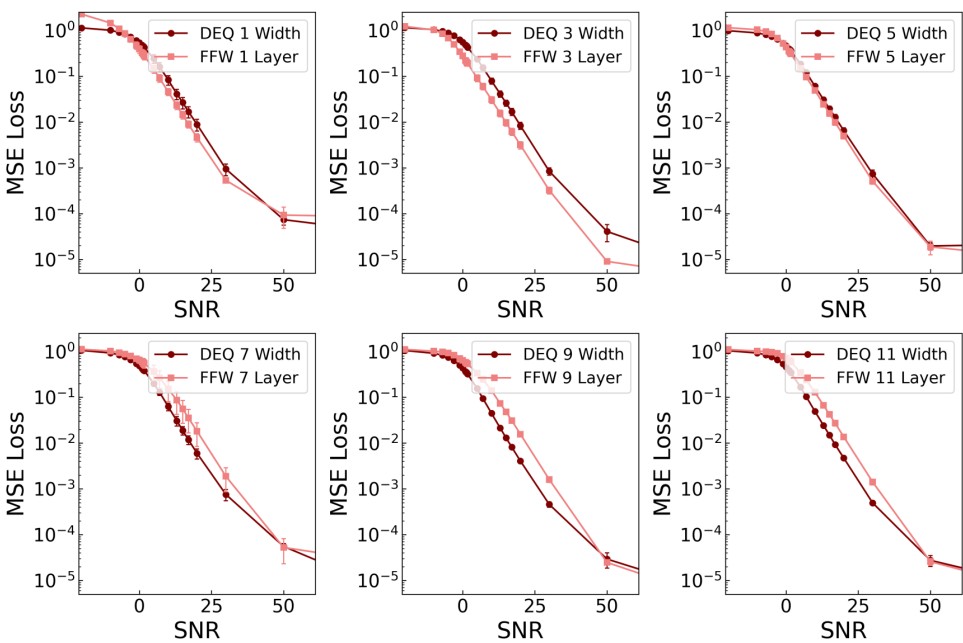

Figure 9: MSE loss for 1-D regression task versus test-time signal SNR in DEQ and FFWs, where the DEQ width is matched to the number of layers in the FFW to equalize the parameters. The input projection layer has a ReLU activation.

The SNR sweeps for each activation are plotted in Figures 7, 8, and 9. We note that the robustness of the FFW models is particularly sensitive to the choice of input projection activation in shallower networks. For example, with ReLU, since the derivative of the activation is zero for negative values, shallower FFW models preserve robustness to noise.

In comparison to classification tasks, the advantage of the DEQ becomes apparent even with shallower FFW architectures. We believe this difference arises because, in classification, there is a range of perturbation that can be tolerated before misclassification occurs, whereas in regression, perturbations directly affect the accuracy of the function being learned.

## D.2 MNIST classification

The accuracy of the DEQ and FFW models on the MNIST classification tasks, as discussed in Section 4.1, is plotted against test-time SNR ranges for three scenarios: no train-time noise (Fig. 10), train-time noise with an SNR of 10dB (Fig. 11), and an SNR of 0dB (Fig. 12). It is noteworthy that the DEQ model exhibits greater robustness to noise compared to the FFW model, especially as the FFW model becomes deeper and at lower SNR ranges.

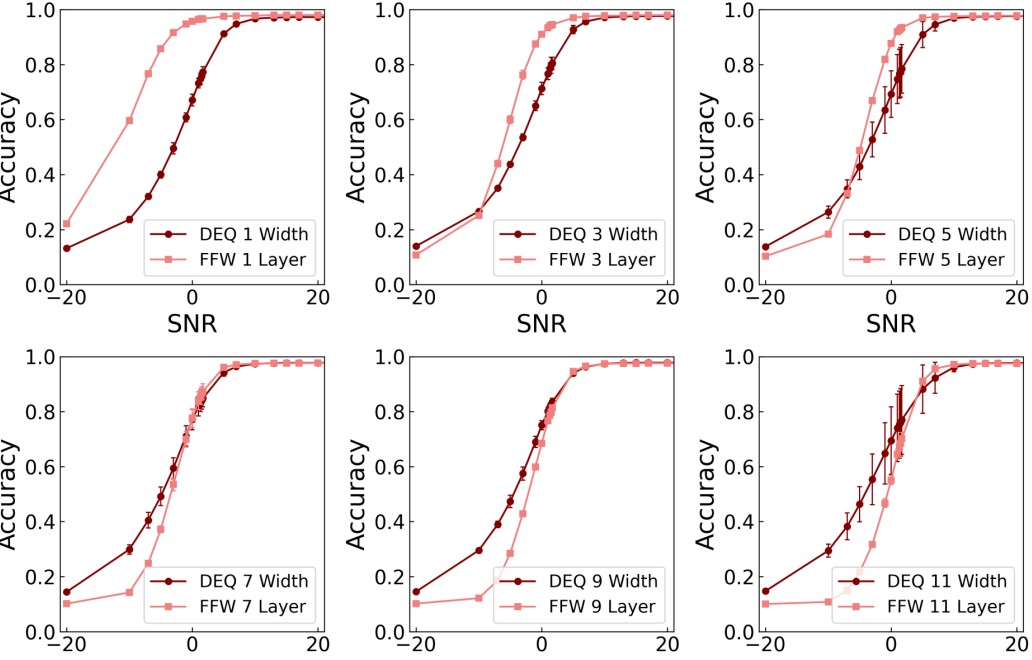

Figure 10: Accuracy versus various test-time signal SNRs in DEQ and FFWs for MNSIT classification trained without noise (SNR= $\infty$), where the DEQ width is matched to the number of layers in the FFW to equalize the parameters.

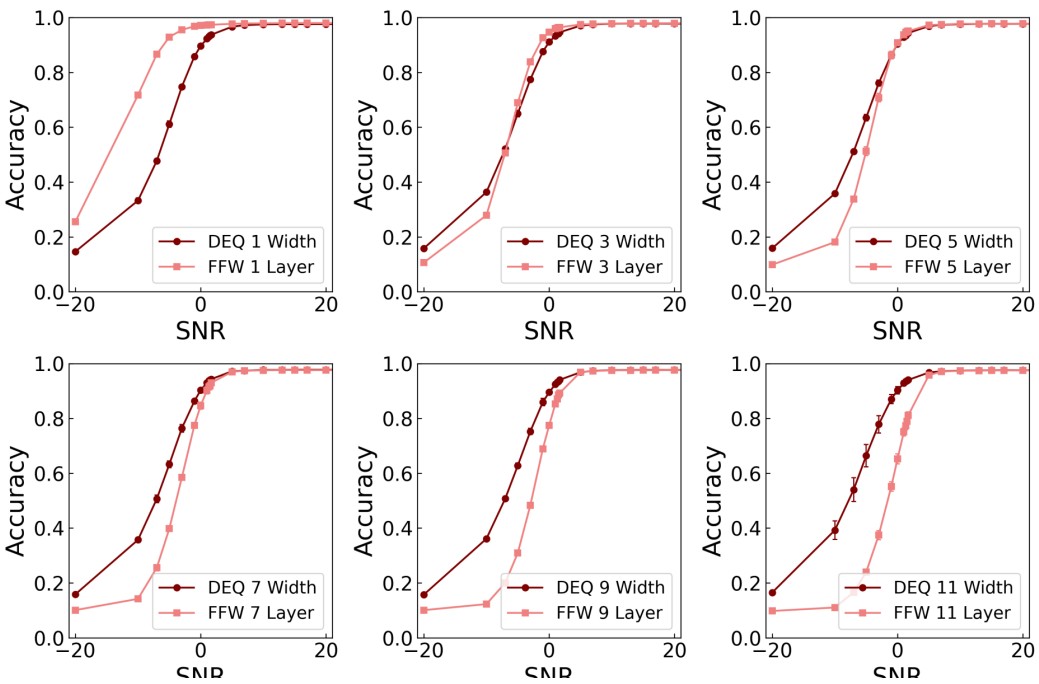

Figure 11: Accuracy versus various test-time signal SNRs in DEQ and FFWs for MNSIT classification trained with noise at SNR= 10dB, where the DEQ width is matched to the number of layers in the FFW to equalize the parameters.

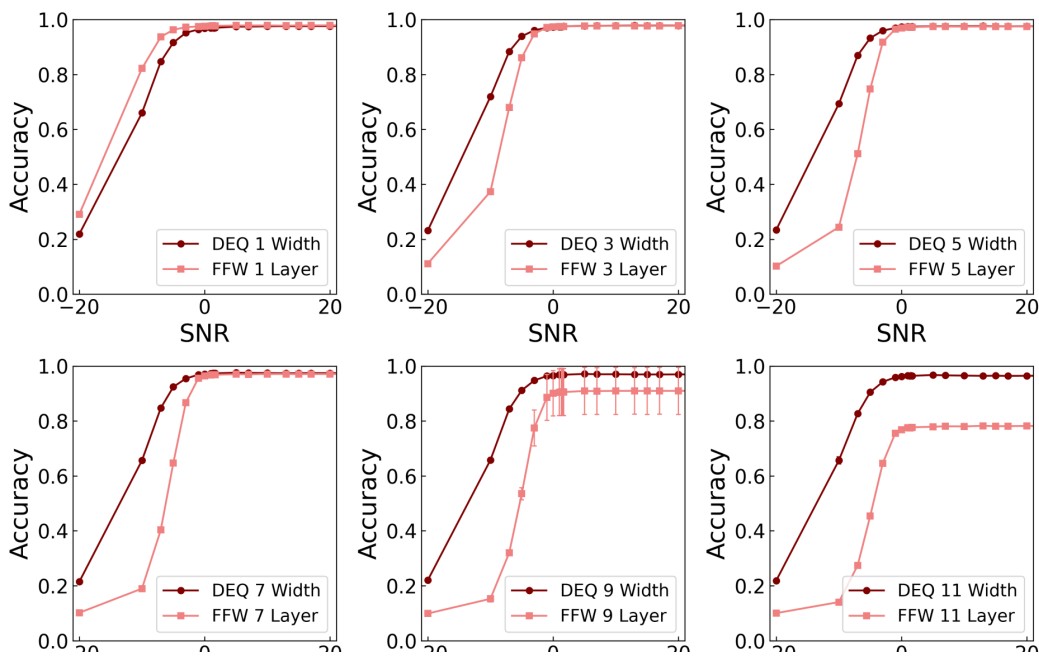

Figure 12: Accuracy versus various test-time signal SNRs in DEQ and FFWs for MNSIT classification trained with noise at SNR= 0dB, where the DEQ width is matched to the number of layers in the FFW to equalize the parameters.

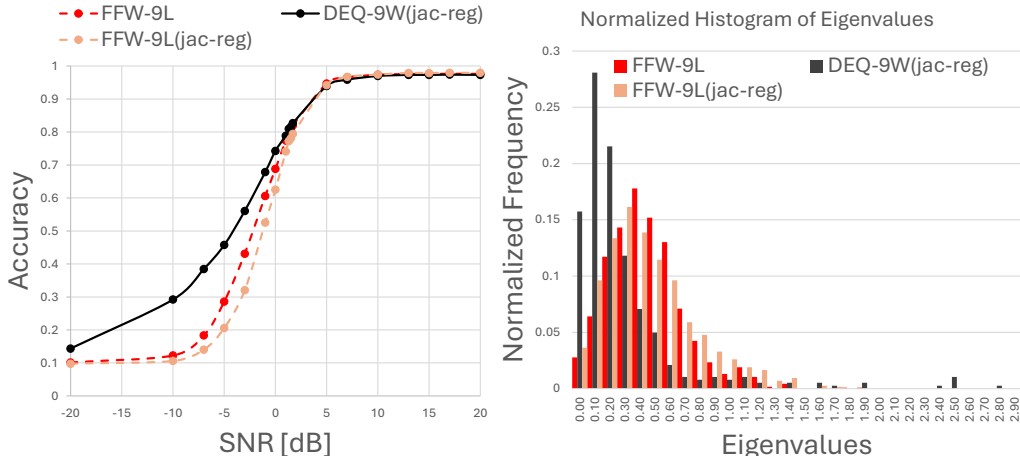

Figure 13: **(Left)** Accuracy versus various test-time SNRs in DEQ (equivalent to 9 layers of FFW in width) and FFW (9 layers) with (coefficient 1 additive with task loss) and without Jacobian regularization. **(Right)** Histogram of the eigenvalues of the weight matrices for FFW and DEQ.

## D.3 Adversarial robustness

We test additional noise defense mechanisms against adversarial attacks on the MNIST dataset. Figures 14 and 15 show the accuracy of DEQ and FFW models under PGD attacks (epsilon representing the strength of the perturbation in $l_\infty$, and the number of steps) as well as AutoAttack (AA), speckle, and salt-and-pepper attacks, alongside various regularization defenses.

Furthermore, we present the effectiveness of the best noise defense for DEQs and FFWs trained with PGD (epsilon=0.3, step-size=0.1, and number of steps=10) in Fig. 16, and for those trained with TRADES (with a loss coefficient of 6, as detailed in [31]) in Fig. 17.

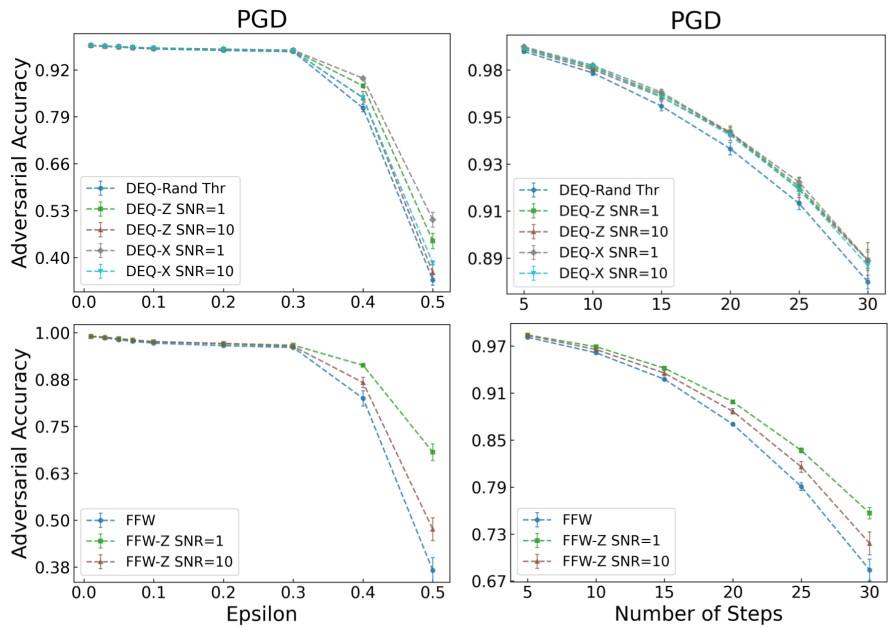

Figure 14: **(Left)** Adversarial accuracy of DEQ (top) and FFWs (bottom) on MNIST classification, regularized with various SNRs and a random thresholding method, against PGD attacks (step size = 0.1) plotted as a function of epsilon (the maximum allowable perturbation under the $l_\infty$ norm). **(Right)** Same parameters as on the left, but plotting adversarial accuracy against the number of steps in the PGD attack.

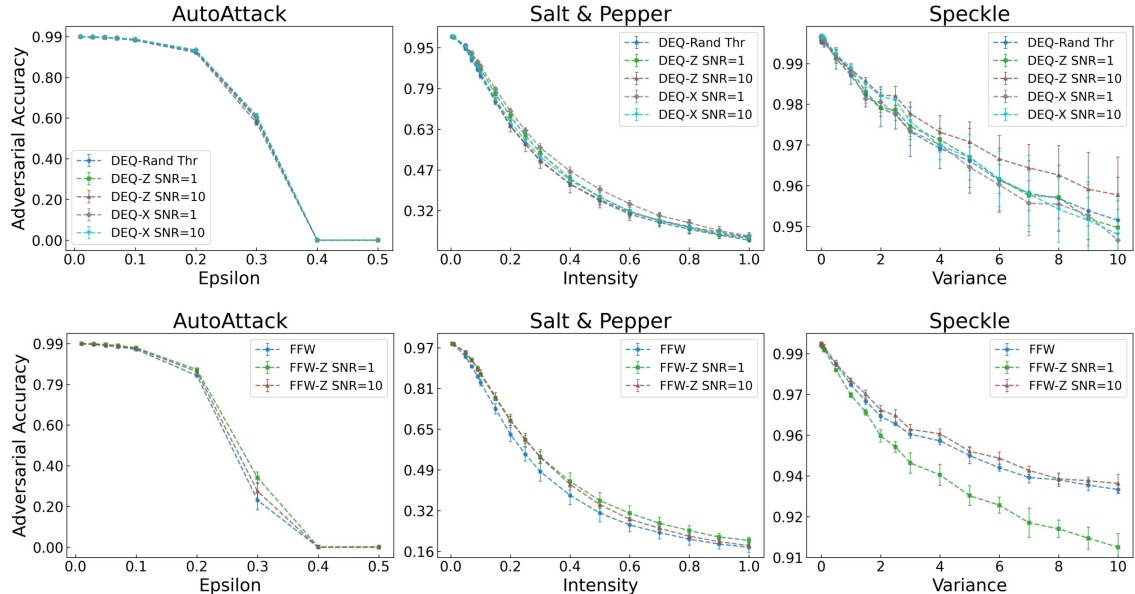

Figure 15: Adversarial accuracy of DEQ **(top)** and FFWs **(bottom)** on MNIST classification, regularized with various SNRs and a random thresholding method, against various attacks. In each plot, the x-axis corresponds to increasing the magnitude of the attack.

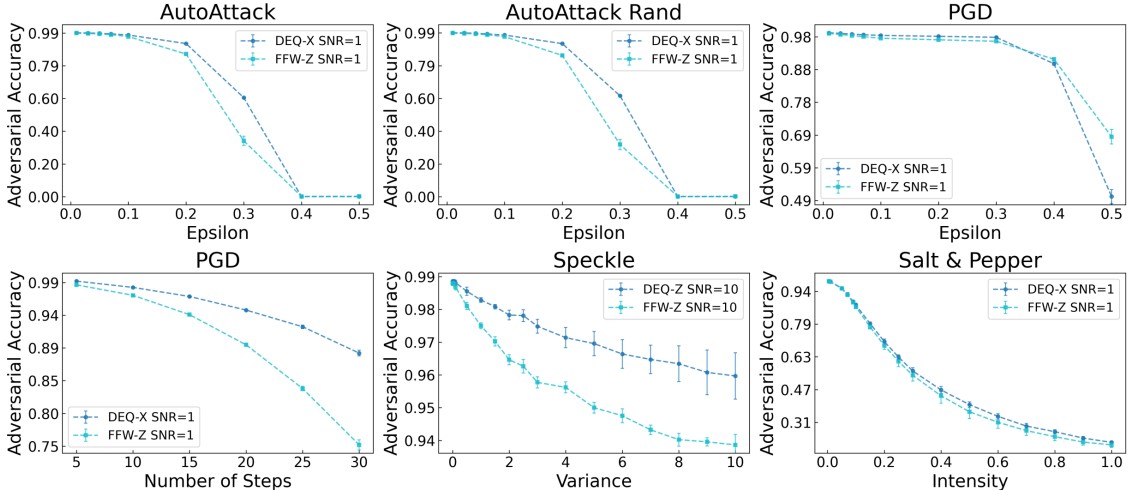

Figure 16: Adversarial accuracy of DEQ and FFW on MNIST classification with the strongest defense and PGD AT against various white-box attacks (AA, AA-Random, PGD-Epsilon, PGD-Number-of-Steps) and black-box attacks (Speckle, Salt&Pepper). In each figure, the leftmost part corresponds to weak attacks (clean accuracy) progressing to strong attacks based on parameters specific to each attack

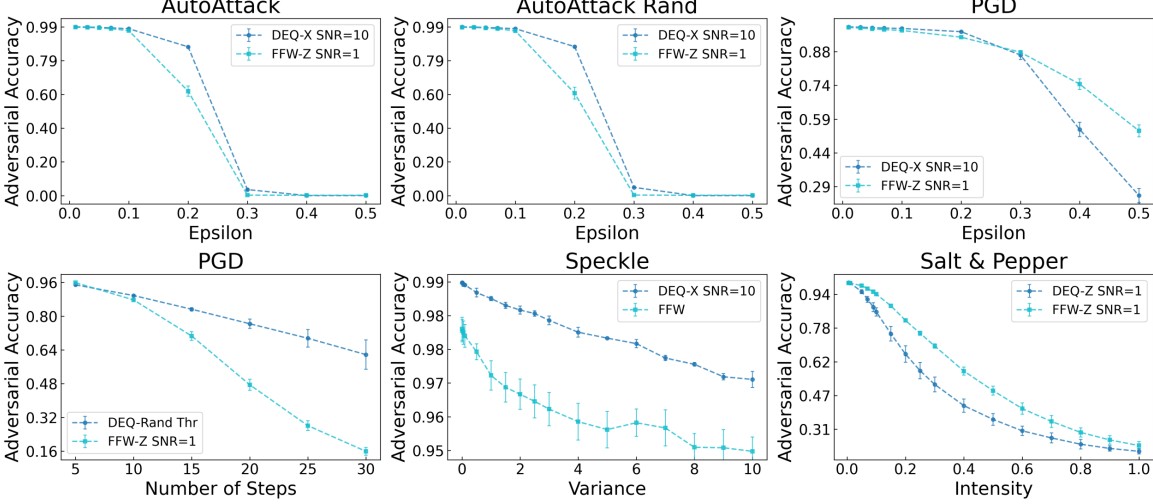

Figure 17: Adversarial accuracy of DEQ and FFW on MNIST classification with the strongest defense and TRADES AT against various white-box attacks (AA, AA-Random, PGD-Epsilon, PGD-Number-of-Steps) and black-box attacks (Speckle, Salt&Pepper). In each figure, the leftmost part corresponds to weak attacks (clean accuracy) progressing to strong attacks based on parameters specific to each attack.

### D.4 Algorithmic extrapolation

We have also compared the performance of FFW and DEQ networks in algorithmic OOD scenarios under noise perturbation. We evaluated the DEQ network's performance on Prefixsum, Maze, and Matrix Addition. In each case, the networks were tested with an extrapolated length (Prefixsum), size (Maze), or distribution shift (Addition). Models are trained and tested under no perturbation, and were trained and tested with perturbation to $z$ with an SNR of $0$ dB. We note that for Prefixsum task, both models suffer under noise. This is perhaps due to the fact that $z$ stores the carry of the binary sums in the $z$ variable of DEQ that is affected under severe noise. For the other two tasks, DEQ is more robust to noise than FFW.

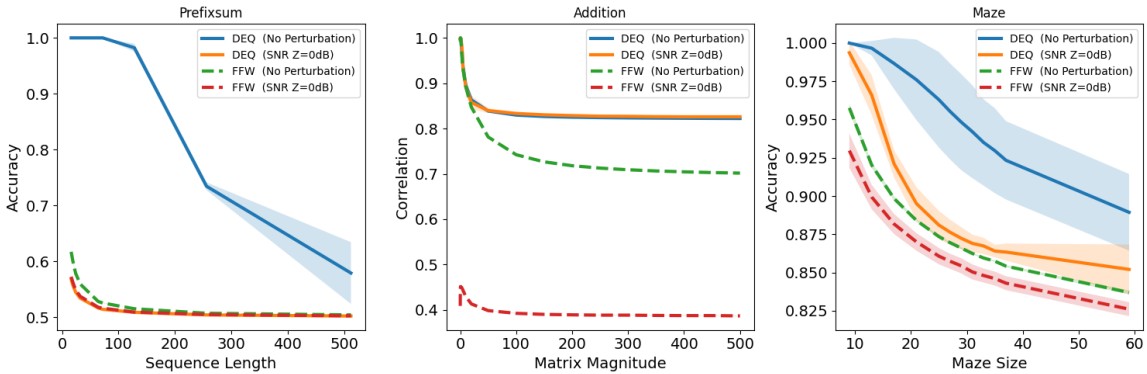

Figure 18: OOD extrapolation and distribution shift algorithmic tasks of FFW and DEQ networks under unperturbed and perturbed models with $z$ SNR of 0dB in **(a)** Prefixsum, **(b)** Maze, **(c)** and Matrix Addition.

## E   Reproducing Experimental Results

In this section, we report the details of experiments to train our neural network models and the corresponding architectures. In all experiments, results are averaged over three seeds.

### E.1   Classification tasks

**For MNIST classification**, the DEQ and FFW models utilize a fully connected architecture. For multilayer FFW, we increased the number of layers with new weights while keeping the hidden size 128 the same in each layer as. In contrast, for DEQ, we increased the width of the hidden state to maintain the number of parameters in both models the same. This adjustment leads to DEQ widths of 128, 223, 288, 340, 386, 427, and 467 corresponding to single-, three-, five-layer, up to thirteen-layer FFW, respectively (the bias parameter count of FFW layers was ignored when computing the width size in DEQ). We used Tanh as the nonlinearity between layers of FFW and iterations of DEQ. The unit-cell of a DEQ or a layer of an FFW then assumes the form:

$$z^{[m+1]} = \text{Tanh}(\boldsymbol{W} z^{[m]} + \boldsymbol{b} + \tilde{\boldsymbol{x}}) \tag{57}$$

We note that both FFW and DEQ have the same input and output projection layers. The input projection layer has a RelU nonlinearity. Details of the network, including training parameters, are presented in Tables 1 and 2.

For MNIST classification, we perturb $\boldsymbol{W}\boldsymbol{z} + \boldsymbol{b} + \tilde{x}$ in Eq. 57, which we refer to as signal perturbation. We employed multiplicative noise and adjusted the signal-to-noise ratio (SNR) level using a coefficient. During training, we used SNR values of either 0dB, 10dB, or no perturbation (an SNR of $\infty$). At inference, we varied the SNR coefficient to evaluate the model under different SNRs.

**For CIFAR-10 classification**, we employed the multiscale DEQ architecture from [4], which processes the image at various scales in branches of the network. We used up to four scales with hidden channel sizes of 32, 64, and 128 for the DEQ architecture. The FFW architecture has a consistent hidden channel size of 128 across all layers. Tables 3 and 4 include details of the architectures and training. For CIFAR-10 classification, we perturb either the $x$, $z$, or activations before nonlinearity in the unit-cell of DEQ or layer of FFW, which we refer to as signal perturbation.

### E.2 Regression tasks

The network architecture of DEQ and FFW for regression tasks is similar to the MNIST classification, except that the input projection layer for regression, is either RelU, SiLU, or identity nonlinearity.

Table 1: MNIST DEQ and 1-D Regression Model Architecture and Training Details. Widths are 128, 223, 288, 340, 386, 427, and 467. * denotes param for training with perturbation.

| Architecture | |
|---|---|
| Input Projection | 784 units, ReLU activation |
| Hidden Layers | width, Tanh activation |
| Output Projection | 10 units, Softmax activation |
| **Training Details** | |
| Epochs | 30 |
| Optimizer | Adam |
| Learning Rate | 0.001 |
| Weight Decay | 0 |
| Jacobian Regularization | Weight: 1.0, Stop Epoch: 55 |
| Train Solver | Name: unrolling, Threshold: 60, Epsilon: 0.001 (0.05*), Stop: rel (mean_rel*) |
| Test Solver | Name: unrolling, Threshold: 60, Epsilon: 0.05, Stop: mean_rel |
| Weight Normalization | False |
| Batch Size | 16 |

Table 2: MNIST FFW and 1-D Regression Model Architecture and Training Details

| Architecture | |
|---|---|
| Input Projection | 784 units, ReLU activation |
| Hidden Layers | $n$ layers, 128 units each, Tanh activation |
| Output Projection | 10 units, Softmax activation |
| **Training Details** | |
| Epochs | 64 |
| Optimizer | Adam |
| Learning Rate | 0.001 |
| Weight Decay | 0 |
| Weight Normalization | False |
| Batch Size | 16 |

Table 3: CIFAR-10 MDEQ Classifier Model Architecture and Training Details

| Architecture | |
|---|---|
| Weight Normalization | True |
| Number of Scales | 4 |
| Scale Channels | [32, 64, 128, 128] |
| Head Channels | [32, 64, 128, 128] |
| Final Channel Size | 400 |
| Batch Norm Momentum | 0.1 |
| Kernel Size | 3 |
| **Training Details** | |
| Batch Size (Train/Test) | 24/24 |
| Epochs | 100 |
| Pretraining Epochs | 9 |
| Optimizer | Adam |
| Learning Rate | 0.001 |
| Weight Decay | 0 |
| Learning Rate Scheduler | CosineAnnealingLR |
| Jacobian Regularization | Weight: 0.4, Stop Epoch: 85 |
| Solver (Train/Test) | Broyden/Broyden |
| Solver Config | Threshold: 40, Epsilon: 0.0001/0.01, Stop Mode: rel/mean_rel |

Table 4: CIFAR-10 FFW Conv2D Classifier Model Architecture and Training Details

| Architecture | |
|---|---|
| Hidden Channels | 128 |
| Final Channel Size | 400 |
| Batch Norm Momentum | 0.1 |
| Kernel Size | 3 |
| Number of Cells per Block | $n$ |
| Weight Normalization | True |
| **Training Details** | |
| Batch Size (Train/Test) | 24/24 |
| Epochs | 100 |
| Optimizer | Adam |
| Learning Rate | 0.001 |
| Weight Decay | 0 |
| Learning Rate Scheduler | CosineAnnealingLR |

### E.3 Adversarial Experiments

We trained the models using either PGD-10 or TRADES. We employed a convolutional architecture comprising of a single conv layer. The same architecture was used for both DEQ and FFW models. During training, in addition to adversarial training, defense mechanism includes perturbation applied to $x$, $z$, or using a random solver's threshold setting, which assigns a threshold that is uniformly set from 1 to $M = 8$. At inference, we utilized a range of white-box or black-box attacks, such as PGD, AutoAttack, AutoAttack-Random, salt-and-pepper, and speckle noise. If a model was trained with an SNR during training, the same SNR was used at inference. The architectures of the models and training details are included in Tables 5 and 6.

Table 5: Adversarial MNIST DEQ Conv2D Classifier Model Architecture and Training Details

| Architecture | |
|---|---|
| Weight Normalization | True |
| Hidden Channels | 32 |
| Number of Cells per Block | 1 |
| **Training Details** | |
| Batch Size (Train/Test) | 32/100 |
| Epochs | 64 |
| Pretraining Epochs | 0 |
| Optimizer | Adam |
| Learning Rate | 0.001 |
| Weight Decay | 0 |
| AMSGrad | True |
| Learning Rate Scheduler | MultiStepLR |
| Milestones | [45] |
| Gamma | 0.1 |
| Solver (Train/Test) | Broyden/Phantom |
| Solver Config (Train/Test) | Threshold: 8/9, Stop Mode: rel/weighttied |

Table 6: Adversarial MNIST FFW Conv2D Classifier Model Architecture and Training Details

| Architecture | |
|---|---|
| Weight Normalization | True |
| Hidden Channels | 32 |
| Number of Cells per Block | 1 |
| **Training Details** | |
| Batch Size (Train/Test) | 32/32 |
| Epochs | 64 |
| Optimizer | Adam |
| Learning Rate | 0.001 |
| Weight Decay | 0 |
| AMSGrad | True |
| Learning Rate Scheduler | MultiStepLR |
| Milestones | [45] |
| Gamma | 0.1 |

### E.4  Algorithmic Tasks

**For the Arithmetic Addition Task**, the dataset [11] is designed to evaluate a neural network's ability to perform numerical addition. The dataset consists of 60,000 data points, with each point comprising two sets of uniformly distributed random values within a defined scale. Training instances use a default scale of 1.0, whereas test instances include a wide range of scales—ranging from 0.1 to 500. The DEQ architecture for this task includes an input dimensionality of 800 (corresponding to two 20x20 matrices to be added), three hidden layers each with 512 units, and an output prediction dimensionality of 400, with Tanh activation functions. No activation function is used for the input projection. The FFW model has the same architecture as the DEQ, with a ReLU activation function for the input projection layer.

**For the Prefix-sum Task**, the dataset [2] contains sequences of binary digits (0s and 1s) used to train a network on sequence-to-classes prediction. The objective of the task is for the network to predict the cumulative parity (even or odd number of 1s) up to the current bit in the sequence. Specifically, the training set consists of binary strings with a fixed size of 32 bits. The model's generalization is then tested on binary string lengths of 16, 18, 20, 24, 32, 64, 72, 128, 256, and 512 bits. For both DEQ and FFW, we use a unit-cell comprising one conv1d with 6 hidden channels and a kernel size of 3, followed by a Tanh nonlinearity.

**For the Mazes Task**, the dataset [2] presents an image-to-classes problem where the network learns to classify the correct path. The training set includes 9x9 mazes, while the test set increases in

complexity with mazes of various sizes, specifically 9x9, 13x13, 17x17, 21x21, 25x25, 27x27, 29x29, 31x31, 33x33, 35x35, 37x37, and 59x59. We follow the training procedure outlined in [2]. For DEQ, we use the residual block structure with two convolutions, each with 128 hidden channels. We do not use any normalization and apply a Tanh nonlinearity. For FFW, we use a single hidden layer residual block with the same nonlinearity as DEQ.

