# OpenReview forum: "Regularizing the Infinite: Improved Generalization Performance with Deep Equilibrium Models"
_NeurIPS.cc/2024/Workshop/MLNCP — MLNCP Poster_

### Official Review · Reviewer_4f61 · 2024-10-05
**Interesting regularisation method for Deep Equilibrium (DEQ) Models**

**Rating:** 7
**Confidence:** 4

**Review:**

This paper studies the use of injecting noise into the activations (intermediate network states) of DEQs. This is a method motivated by previous findings that such noise injection made RNNs more robust to input perturbations. The proposed method is a straightforward application of this idea and is evaluated on several tasks and benchmarks. Overall, the work is novel and a good fit to the workshop.

**Strengths:**
- Interesting method that seems to work well at improving robustness to noisy inputs.
- I appreciated the attempt to systematically characterise when exactly DEQs are more robust than feedforward (FFW) networks with the same parameter count.

**Limitations:**
- I did not find the theoretical arguments comparing DEQs and FFW models very compelling. The derivation of the network sensitivity is straightforward enough, but the argued implication that DEQs are more robust only in comparison to deeper FFW models does not necessarily follow from the presented analysis. This seems to be more of an empirical finding, and I would suggest the authors to rephrase/soften their statements in the introduction and abstract accordingly.
- The use of Gaussian noise in the inputs seems to me a very limited method to test robustness. For example, is it really surprising that a model trained with noise is more robust to the same kinds of noise? Some comments from the authors on this point would be appreciated.

---

### Decision · Program_Chairs · 2024-10-10

Accept (Poster)